# Behavioral and Electrophysiological Arguments in Favor of a Relationship between Impulsivity, Risk-Taking, and Success on the Iowa Gambling Task

**DOI:** 10.3390/brainsci9100248

**Published:** 2019-09-24

**Authors:** Julie Giustiniani, Coralie Joucla, Djamila Bennabi, Magali Nicolier, Thibault Chabin, Caroline Masse, Benoît Trojak, Pierre Vandel, Emmanuel Haffen, Damien Gabriel

**Affiliations:** 1Department of Clinical Psychiatry, University Hospital of Besançon, 25000 Besançon, France; djamila.bennabi@univ-fcomte.fr (D.B.); mnicolier@chu-besancon.fr (M.N.); cmasse@chu-besancon.fr (C.M.); pierre.vandel@univ-fcomte.fr (P.V.); emmanuel.haffen@univ-fcomte.fr (E.H.); 2Laboratory of Neurosciences, University of Burgundy Franche-Comté, 25000 Besançon, France; coralie.joucla.pro@gmail.com (C.J.); tchabin@edu.univ-fcomte.fr (T.C.); dgabriel@chu-besancon.fr (D.G.); 3Clinical Investigation Centre, University Hospital of Besançon, Inserm, CIC 1431, 25000 Besançon, France; 4FondaMental Foundation, 94000 Créteil, France; 5Department of Psychiatry and Addictology, University Hospital of Dijon, 21079 Dijon, France; benoit.trojak@chu-dijon.fr; 6EA 4452, LPPM, University of Burgundy Franche-Comté, 21000 Dijon, France

**Keywords:** decision-making, IGT, BART, impulsivity, risk-taking, theta oscillations

## Abstract

The aim of the present study was to investigate the relationship between trait impulsivity, risk-taking, and decision-making performance. We recruited 20 healthy participants who performed the Iowa Gambling Task (IGT) and the Balloon Analog Risk Task (BART) to measure decision-making and risk-taking. The impulsivity was measured by the Barratt Impulsiveness Scale. Resting-state neural activity was recorded to explore whether brain oscillatory rhythms provide important information about the dispositional trait of impulsivity. We found a significant correlation between the ability to develop a successful strategy and the propensity to take more risks in the first trials of the BART. Risk-taking was negatively correlated with cognitive impulsivity in participants who were unable to develop a successful strategy. Neither risk-taking nor decision-making was correlated with cortical asymmetry. In a more exploratory approach, the group was sub-divided in function of participants’ performances at the IGT. We found that the group who developed a successful strategy at the IGT was more prone to risk, whereas the group who failed showed a greater cognitive impulsivity. These results emphasize the need for individuals to explore their environment to develop a successful strategy in uncertain situations, which may not be possible without taking risks.

## 1. Introduction

Decision-making is a complex cognitive process that is used, in part, to solve open and risky problems in daily life, the outcomes of which are often unpredictable [1,2]. Impairments in decision-making can have harmful direct consequences on the social and personal aspects of one’s daily life. Such impairment has been observed in many neuropsychiatric disorders [3,4,5], such as behavioral addictions [6] and substance addictions [7,8,9].

The Iowa Gambling Task (IGT) is a psychological task designed to assess impairments in decision-making under conditions of uncertainty by simulating real-life economic decisions [10]. The IGT is principally used to describe neurophysiological and clinical differences between pathological and healthy populations [7,8,9,10,11], however, within-group differences are frequent and expected on this task. Indeed, several studies have reported a failure rate of up to 55% on the IGT in healthy populations for both the original and adapted versions [7,12,13,14,15].

If the IGT was originally designed to support the somatic marker hypothesis [1,16], several other factors appear to have a major impact on decision-making. Since the IGT was founded on the principle that real-life decision-making involves choices that are based on expected but uncertain rewards and penalties, and that optimal choices are based on well-considered strategies, one can suggest that impulse control problems may alter our decisions. Impulsivity appears to be one factor that could explain the differences in IGT performance. Impulsivity is considered an important element in the decision-making process; however, it is a complex construct with several components, making the discernment of its exact role difficult. Impulsivity is nevertheless known to be related to risk-taking and a lack of reflection between environmental stimuli and the behavioral response during decision-making [17]. Impulsivity is often measured by self-report, such as the Barratt Impulsiveness Scale (BIS) [18]. Self-report scales can provide valuable information about impulsivity as a stable trait in individuals [19]. Impulsivity is a trait known for being able to influence decision-making performance [20,21], however, the influence of impulsive personality traits on IGT performance remains unclear [22,23]. Although self-report scales can measure relatively stable characteristics, they are subjective and cannot directly measure the biological foundations of impulsivity. To provide a more objective and biologically based measure, the Balloon Analogue Risk Task (BART) was specifically designed to measure the propensity for risk-taking [24,25]. Impulsivity is indeed closely related to risk-taking. High levels of impulsivity often result in risky behaviors, such as the initiation of alcohol or drug use [26]. At the same time, almost any human decision carries some risk. Decision-making and risk are two inter-related factors, both being related to various uncertainties [27]. Tests of performance on the IGT and BART in subjects with high and low trait impulsivity revealed that trait impulsivity did not lead to different scores on the IGT or BART. Nevertheless, participants with low trait impulsivity were seemingly more risk averse on the BART, although they made more advantageous selections at the IGT [28]. Clarifying how these concepts are entangled appears to be important in both theoretical and clinical aspects. Changes in decisions are frequently justified by an important feeling, led by impulsivity or risk-taking with a lack of distinction of how each of these constructs specifically influences behaviors and decisions. Indeed, the somatic marker hypothesis refers to the collection of signals related to the body and the brain that characterize the emotional and affective responses [29]. In this sense, the impact of impulsivity or risk-taking on decision-making does not contradict the somatic marker hypothesis.

In this context, the aim of the present study was to further investigate the relationship between impulsivity, risk-taking, and IGT performance. We examined whether the ability to develop a correct strategy on the IGT requires specific levels of trait impulsivity and risk-taking on the BART. To provide important additional information about impulsivity as a dispositional trait, we also recorded neural activity at rest. With resting-state electroencephalography (EEG), tonic cortical activity prior to task performance can be measured, reflecting individual behavioral and cognitive disposition, which has shown to be stable for several years [30]. Therefore, baseline anterior asymmetries can be used as a predictor of performance on behavioral tasks [31]. The left prefrontal cortex (PFC) was previously associated to individual predisposition to respond in terms of approach-related and reward-related motivations and emotions [32,33,34,35]. Based on these previous results, we hypothesized that hemispheric asymmetry predicts behavioral profiles with regard to performance on the BART, potentially explaining IGT performance. Exploring such a neural and behavioral relationship may have direct implications on the understanding of mechanisms underlying many neuropsychiatric disorders in which decision-making is altered.

## 2. Materials and Methods

### 2.1. Participants

Twenty healthy, right-handed subjects (10 male and 10 female, mean age = 38.7 years, SD = 18.3 years, range = 21–59 years) participated in the study. Participants had no previous medical history of psychiatric disorders, substance abuse, alcohol abuse, neurological disease, traumatic brain injury, or stroke, and did not take any medication.

The participants received information regarding the aims and procedures of the study and were provided with written informed consent to participate. Because of the influence of real money on motivation, the subjects were informed that the monetary payment would be proportional to their gains in the game, with an exchange rate of 1%. Because of ethical considerations, regardless of their performance, all of the participants received 75€ at the end of the study. The protocol was approved by the Committee of Protection of Persons (CPP-Est II) and was conducted in accordance with the principles of the Declaration of Helsinki (ClinicalTrials.gov identifier: NCT02873572) (for details, see Giustiniani et al., 2015 [14]).

### 2.2. Experimental Tasks

#### 2.2.1. Iowa Gambling Task 

We used an electronic version of the IGT [10], which was adapted to study event-related potentials and to analyze brain activity. The aim of the IGT was to win as much money as possible by making successive selections between four decks [10,36].

A full description of the study is presented elsewhere [14]. Minimal changes were made to adapt the IGT to EEG. First, to extend electrophysiological recordings of the exploration phase, the number of trials was increased from 100 to 200 and the subjects were given no hints about the presence of advantageous or disadvantageous decks. Second, the design of the trials was modified to minimize ocular artifacts. In each trial, subjects were instructed to fix their gaze on a cross or letter while making their selection by pressing a key. After the selection, feedback regarding the deck that was chosen and total credit amount was displayed, followed by the amount of money won in the trial. A fixation point then appeared to orient the eyes to the center of the screen, followed by a fixed letter that announced the result.

Once the task was complete, the subjects’ performance and psychometric test response were analyzed (for details, see Giustiniani et al., 2015 [14]), and sub-groups were constituted. The sub-groups did not differ with regard to age (*p* = 0.21), level of education (*p* = 0.62), or sex (*p* = 0.37). Additionally, no differences were found between groups in various psychometric tests that evaluated neuropsychiatric disorders, addiction, or personality dimensions. Namely, there were no differences on the South Oaks Gambling Screen (SOGS) [37], Alcohol Use Disorder Identification Test AUDIT [38], Diminuer Entourage Trop Cannabis/Cut-down Annoyed Guilty Eye-opener DETC/CAGE [38], Fagerström test [38], Beck Depression Inventory, abbreviated version (BDI) [39], Liebowitz scale [40], or the Big Five Inventory-French (BFI-Fr) [41] (see Table 1). Sub-groups were analyzed in a more exploratory approach, to get more insight into the mechanisms observed on the whole group of participants.

#### 2.2.2. EEG Recording at Rest 

An EEG recording at rest was performed immediately after the IGT. During the recording, subjects were instructed to close their eyes and remain still. Resting-state EEGs were recorded for four minutes using OSG digital equipment (BrainRT, OSG bvba, Rumst, Belgium) with two Schwarzer AHNS epas 44-channel amplifiers (Natus, Munich, Germany). A 64-channel electrode cap (Easycap, easycapGmbh, Ammersee, Germany) with electrodes positioned in the 10/10 system was used to get the EEG signals. EEG data were continuously recorded with a band pass of 0.05–100 Hz and sampling rate of 1000 Hz. All electrode impedances were below 30 kΩ before recording was started.

Two electrodes (AF3 and AF4) were placed over the left and right dorsolateral PFC. Signal processing was performed using EEGlab and Cartool software (http:/brainmapping.unige.ch/Cartool.php). Raw EEG data were digitally low-pass filtered and corrected for eye movements. Raw EEG data were offline re-referenced to a common average. Channels with excessive noise (due to malfunctioning or bad signal during data collection) were replaced using topographic interpolation. EEG signals that contained residual muscle movements or other forms of artifacts at >50 µV were removed prior to further analysis. Each artifact-free period was further segmented into epochs of 2 s A fast-Fourier transform (Hamming window: length 10%) was used to estimate spectral power in the θ (4–7 Hz) and α (8–12 Hz) frequency bands. Finally, frontal brain asymmetry was calculated by subtracting the mean power densities (AF3–AF4) in each frequency band.

#### 2.2.3. Balloon Analog Risk Task

The BART is an informatic-based measure of the propensity for risk-taking [24]. During the BART, the participants were required to sequentially press a button to inflate a series of 30 balloons that were displayed on the computer screen. The computer screen showed a small simulated balloon accompanied by a balloon pump, a reset button labeled “Collect $$$,” and a permanent display of money earned, labeled “Total Earned”. Each pump corresponded to 5 cents that accrued in a temporary reserve (the amount of money in this reserve was never indicated to the participant). At any point during each balloon inflation, the participant could stop pumping the balloon and click the “Collect $$$” button. Clicking this button transferred all of the money from the temporary reserve to the permanent bank, at which time the new total that was earned was incrementally updated cent-by-cent while a slot machine payoff sound effect played. The balloon could either grow larger or explode. A larger balloon was simultaneously associated with a greater probability of explosion and a larger virtual monetary reward. Once a balloon was pumped past its individual explosion point, it exploded with a “pop” sound, at which time all of the money in the temporary reserve was then lost. After each balloon explosion or money collection, the participant’s exposure to that balloon ended and a new balloon appeared, until all 30 balloons had been displayed. All of the balloons had a different explosion threshold. Balloon breakpoints ranged from 1 to 128 pumps. The participants were instructed to maximize the amount of virtual reward during the experiment. Risk-taking behavior on the BART was measured by calculating the total and mean adjusted number of pumps across balloons (only trials in which the balloons did not explode were included in the calculations).

### 2.3. Data Analyses

To compare the IGT with other parameters (trait impulsivity, BART scores, and frontal asymmetry), the individual net score was calculated by subtracting the number of disadvantageous decks from the number of advantageous decks that were selected in the last 60 trials. We chose not to focus on the total net score because some studies have proposed that performance at the IGT is more reliable when participants have developed a certain strategy [43]. The last 60 trials was the chosen cutoff point at which we considered that participants had started developing a strategy because the net score for the last 60 trials was significantly higher than the net score in the first trials (see Giustiniani et al., 2015 for more details [14]). The subjects were classified into two groups according to their net score on the IGT. Based on classifications set in previous studies [7,10,29], nine subjects were classified as favorable (net score > 10) and 11 subjects were classified as undecided (net score = 10 to −10). No subject had a net score less than −10.

Trait impulsivity was measured by the BIS-10 [18,42] and included total and subscale scores (cognitive, motor, and non-planning). There were two parameters of the BART considered for analysis, namely total number of adjusted pumps, which was directly related to money earned at the end of the experiment (each adjusted pump corresponded to 5 cents earned), and average number of adjusted pumps, which was calculated by dividing the number of completed trials by the total number of adjusted pumps. As a greater average suggests a greater inclination of participants to take risks, the second parameter is more related to risk-taking.

The performance on the BART as it pertains to trials was taken into account by analyzing the task in three blocks with 10 trials each [24,25]. To determine if data from the favorable and undecided groups had a common mean during throughout their participation in the BART, repeated-measures analysis of variance (ANOVA) were performed. The threshold of significance was set to 5% and post hoc analyses were performed using Fischer’s Least Significant Difference (LSD) test.

Relationships between net score on the IGT, trait impulsivity, risk-taking, and frontal asymmetry were assessed using nonparametric Spearman rank-order correlations. To consider multiple comparisons, the threshold of significance was set to 1%. We performed the analysis using Statistica 11.0 for Windows (StatSoft, Inc., Tulsa, OK).

Given significant relationships for the participants as a whole, additional correlations were then evaluated specifically in the favorable and undecided groups.

## 3. Results

### 3.1. Relationship between the Participants’ Strategy on the IGT and Risk-Taking on the BART 

We investigated whether risk-taking would influence performances on the IGT by studying the relationship between IGT scores and presses on the BART. We observed that both the average adjusted number of pumps and the total number of pumps were reduced in the first ten trials of the BART compared to the other blocks. A repeated-measures analysis of variance (ANOVA), with block (trials 1–10, trials 11–20, trials 21–30) as factor, confirmed these observations by showing significant differences on the average adjusted number of pumps during the different blocks (*F*_2,38_ = 10.9982, *p* < 0.001), with reduced performances in trials 1–10 compared to 11–20 (*p* < 0.05, Bonferroni corrected) and 21–30 (*p* < 0.001, Bonferroni corrected).

To study whether the reduced performances in the first 10 trials could be imputable to the development of an efficient strategy, we explored their relationship with IGT performances. The first trials of the BART display an uncertainty level which confers a structure close to the last trials of the IGT [44]. A significant correlation was found between the net score on the IGT during the last 60 trials and the total adjusted number of pumps in the first 10 trials of the BART (Spearman *R* = 0.48, *t*_18_ = 2.38, *p* = 0.03; Figure 1). No such relationship was found between the total net score nor the net score during the first 40 trials and the average adjusted number of pumps. Furthermore, no relationship was found between the IGT net score during the last 60 trials of the IGT and the total adjusted number of pumps in the trials 11–20 (Spearman *R* = −0.03, *t*_18_ = −0.15, *p* = 0.88) and 21–30 (Spearman *R* = −0.02, *t*_18_ = −0.07, *p* = 0.94) of the BART.

To explore further the observations made on the whole group of participants, we investigated whether risk-taking at the BART was more pronounced in participants able to develop a successful strategy. The data were analyzed using repeated-measures ANOVA, with block (trials 1–10, trials 11–20, trials 21–30) and group (favorable, undecided) as factors. A significant block x group interaction (F_2,36_ = 6.2811, *p* < 0.01; Figure 2) was observed. The group who failed to develop a favorable strategy showed less risk-taking, i.e. fewer presses compared to the Favorable group in the first 10 trials of the BART (Fisher LSD test, *p* < 0.05). Risk-taking of the undecided group was also reduced in the first 10 trials compared to trials 11–20 and trials 21–30 (*p* < 0.01 for both comparisons). No relationship between the BART and the IGT was found on the average number of adjusted responses.

### 3.2. Relationship between the Participants’ Strategy on the IGT and Impulsivity

No relationship was found between net score on the IGT and BIS-10 global (Spearman *R* = ‒0.08, *t*_18_ = ‒0.35, *p* = 0.73) as well as the subscale scores. This absence of relationship could also be observed for the favorable and the undecided groups.

### 3.3. Relationship between Impulsivity and Risk-Taking

We also evaluated whether risk-taking could be defined as a component of impulsivity by studying the relationship between impulsivity scales and risk-taking (Table 2). Surprisingly, there was a significant negative correlation between the average number of adjusted pumps on the BART and cognitive impulsivity on the BIS-10 in the whole group (Spearman *R* = −0.68, *t*_18_ = −3.89, *p* < 0.01), suggesting that cognitive impulsivity is associate with a decreased propensity in risk-taking. This negative correlation particularly strong in the undecided group (Spearman *R* = −0.77, *t*_9_ = −3.59, *p* < 0.01) but not in the favorable group (*p* > 0.05). No correlation was found between the total number of adjusted pumps on the BART and the global BIS score and all subscales (all *p*s > 0.05).

### 3.4. Relationship between Cortical Asymmetry and Behavioral Scores

Frontal asymmetry was measured to examine how the dispositional trait of impulsivity could affect risk-taking behavior. No significant relationship was observed between IGT net scores and theta asymmetry (*p* = 0.81) or alpha asymmetry (*p* = 0.12). There was also no relationship between BART, theta (total number of pumps: *p* = 0.83; average number of pumps: *p* = 0.16) and alpha asymmetry (total number of pumps: *p* = 0.37; average number of pumps: *p* = 0.42).

We nevertheless questioned whether a subgroup of subjects could be affected by cortical asymmetry. Nonparametric Spearman rank-order correlation analysis on sub-groups revealed a very strong, significant relationship between the mean adjusted number of balloon pumps and theta asymmetry in favorable subjects (Spearman *R* = 0.9, *t*_7_ = 5.462793, *p* < 0.001). Figure 3 shows the relationship between frontal theta asymmetry and risk-taking behavior. No significant relationship was found for either the alpha frequency range or undecided subjects. Additionally, no significant relationship was found between the BIS-10 global and subscale scores and theta asymmetry (all *p*s > 0.1).

We investigated whether participants who were able to develop the correct strategy on the IGT presented differences in impulsivity compared with undecided subjects. Favorable and undecided participants did not differ in any components of the BIS-10 (all *p*s > 0.1 for global BIS score and all subscales). No significant relationship was found between BIS-10 global and subscale scores and theta asymmetry (*p* > 0.1).

## 4. Discussion

The present study explored the role of impulsivity and risk-taking on decision-making performance on the IGT and the impact of frontal lobe asymmetry on decision-making.

### 4.1. Association between Decision-Making, Risk-Taking, and Impulsivity

Although the sensitivity of the IGT in detecting impairments in decision-making under conditions of uncertainty is well established and numerous studies have highlighted the complexity of this task and challenge this pretense for understanding the functions or dysfunctions it purportedly measures [16,45,46]. A positive correlation was found between the net score on the IGT during the conceptual phase (the last 60 trials) and the propensity to inflate the balloon during the first block of the BART. Differences in impulsivity and risk-taking were found between participants who were able to develop a correct strategy and those who could not; indeed, we found that the undecided group made significantly fewer pumps than the favorable group in the first block of the BART. Differences in impulsivity and risk-taking were found between participants who were able to develop a correct strategy and those who could not. The undecided group made significantly fewer pumps than the favorable group in the first block of the BART. The undecided group was more cautious than the favorable group at the beginning of the task, even if both groups had the same level of performances in the following blocks. Compared to the undecided group, favorable group took more risk at the beginning of the task, to achieve more rapidly a high level of performance.

To understand this association between the later stage of the IGT and BART scores, risk-taking in the early and late stages of the IGT should be considered separately [45]. This is different from the BART, which was specifically designed to measure the propensity for risk-taking from the beginning of the task [24,25]. Nevertheless, the correlation observed between the IGT and the first block of the BART is not inconsistent with the literature. During the BART, participants did not know about the probability distribution of explosions which confers an uncertainty level at the task in addition to the risk, mostly on the first trials [44]. In this context, the first trials of the BART share similar mechanisms as the last trials of the IGT. In contrast to the early stages of the IGT, during which risk-taking is not a deliberate act but rather reflects a failure to recognize risk, risks are explicit in the later stages of the IGT. Thus, the propensity for risk-taking can be measured only at the end of the IGT and only if the subject develops explicit knowledge of the risk profile [28].

Participants who make advantageous decisions in the later trials of the IGT are those who have developed explicit knowledge of the risky decks and the development of such knowledge may be more related to the propensity for risk-taking than trait impulsivity [28]. Our initial hypothesis that we would observe increased risk-taking in the undecided group was not confirmed. It was actually quite the opposite that was observed, the undecided group showing a reduced risk propensity at the beginning of the BART. The subjects in the undecided group were unable to develop the appropriate strategies and consequently did not consciously take risks [28]. Low impulsivity has been previously shown to be correlated with better performance on the IGT and fewer pumps on the BART [28]. Converse to the previous study, in which the groups were formed based on the subjects’ impulsivity, group assignment in the present study was based on IGT performance. Furthermore, we did not find a link between the net score on the IGT and impulsivity. This suggests that impulsivity alone cannot explain decision-making abilities, suggesting a complex process. The difference observed in the total number of adjusted pumps in the first block suggests that the favorable group took more risk at the beginning of the BART. These risks accelerated the learning process, thereby allowing the favorable group to earn more money at the start of the game. Since subjects of the undecided group showed a reduced propensity to take risk in the different tasks, they consequently learned more slowly the probability of explosions at the BART and decks composition at the IGT. More risk-taking is therefore not necessary related to a pathological process and, in this case, a lower level of risk-taking could lead to bad decision-making. Impact of risk-taking during the IGT appears to be more complex than only unidirectional.

### 4.2. Association between Impulsivity and Risk-Taking

We found a strong, negative relationship between a high impulsivity level as rated by the cognitive subscale of the BIS-10 and risk-taking. When we continued analysis on sub-group, we found this correlation in the undecided group, but not in the favorable group. The cognitive subscale is related to attentional impulsiveness (i.e., difficulty maintaining attention to relevant details) [47]. Although no correlation was found between impulsivity and IGT performance, the association between impulsivity and risk-taking in the undecided group suggests an indirect link between cognitive impulsivity and the inability to develop an appropriate strategy on the IGT. The difficulty maintaining attention to relevant details may explain why these subjects were unable to develop a correct strategy on the IGT. The lack of a significant relationship between impulsivity on the BIS-10 and risk-taking on the BART in the favorable group does not necessarily imply that there are no impulsive components to this group. The impulsivity construct proposed by Dickman [48] and the two types of impulsivity: Functional and dysfunctional [49] may help explain why differences were found only in the first block of the BART in the group who developed a correct strategy. Functional impulsivity is related to a tendency to make decisions quickly and is principally relevant in situations with personal gain, whereas dysfunctional impulsivity is related to a lack of reflection, resulting in negative consequences. In the first block of the BART, the subjects in the favorable group presented a stronger propensity to inflate. In the other blocks, the subjects in the undecided group tended to pump more to further inflate the balloon, as opposed to subjects in the favorable group, who adjusted their strategy as a function of the risk they perceived. Subjects with the individual trait of functional impulsivity tended to make impulsive decisions when they perceived that strategy high level of risky behavior would result in positive outcomes [48]. Moreover, functional impulsivity has been associated with Gray’s concept of reward reactivity [50]. Functional impulsivity is positively related to measures of the Behavioral Approach System (BAS) and negatively related to measures of the Behavioral Inhibition System (BIS). Consistent with this concept, subjects with high levels of functional impulsivity appeared to be more motivated by and reactive to situations that likely lead to reward [50]. Risk-taking has often been associated with motivational drives and behavior [51], measured by a high score on the BAS [52]. Here, we propose that risk-taking is impacted by both impulsivity and motivation, because participants are willing to take risks for the potential of the associated reward. Increased motivation in the favorable group may explain why their behavioral performances were improved following each task. The subjects in the favorable group may have exhibited functional impulsivity and greater motivation to explore the game, therefore leading to development of the correct strategy.

### 4.3. Association between Neural Activity and Behavioral Performance

The neural response at rest supports the positive influence of risk-taking on behavior to a certain extent. Consistent with a previous study, we found that individual differences in risky choice behavior could be predicted by the hemispheric balance of resting-state PFC activity [30,53,54]. Higher theta-band power detected by the left electrodes was associated with a greater mean adjusted number of pumps in the favorable group. A negative correlation was previously reported between resting-state theta oscillations and glucose metabolism, in which theta oscillations at rest indicated a low level of neural activity [54]. Right theta oscillations may reflect right hypoactivity, which has been associated with a lack of regulatory ability to suppress a more seductive choice, due to immediate higher payoffs [30]. The right PFC is involved in withdrawal-related emotions [55,56] and a logical assumption is that hypoactivity in the right PFC is negatively correlated with BIS scores. We may postulate that left hemisphere hypoactivity reflects activity of the right hemisphere. Thus, the favorable group should have had better regulatory abilities to suppress immediate appealing options. Thus, subjects who can develop a successful strategy may, therefore, be likely to have lesser left prefrontal activity, favoring right hemisphere activity, which may indicate a better ability of this group to avoid more immediately attractive options with negative consequences in the long term.

An alternative hypothesis is that theta oscillations are also involved in learning and emotion regulation and may be involved in the encoding of information, particularly during active exploratory phases [57]. This specific theta activity shows that the quality of emotional regulation impacts performance. Theta activity has been associated with emotion regulation and encoding information, with both elements being particularly important for the somatic marker hypothesis, which states that emotions affect information encoding and the decision-making process. According to theories of emotion and motivation, the right PFC is related to negative emotions and drives withdrawal behavior, whereas the left PFC is related to positive emotions and drives approach behavior [34,58]. Prior studies have demonstrated a negative correlation with BIS scores and right theta oscillations [54]. This cortical asymmetry is a well-known correlate of approach motivation and risk-taking [52]. Somatic markers traduced an unconscious and conscious knowledge of the risk. This is reflected by a change in the autonomic arousal before decks selections [1,7]. However, the risk anticipation does not seem to imply strategy differences and differences in risk-taking in healthy subjects [14]. Furthermore, risk-taking requires a conscious or unconscious knowledge of the risk level [44]. In the present study, the participants in the favorable group exhibited a dominant approach behavior and consequently took more risks, resulting in positive consequences for their decision-making under conditions of uncertainty. Finally, absence of a significant correlation between theta asymmetry and net score on the IGT confirms that this theta asymmetry exclusively reflected risk-taking and that understanding the results of IGT performances cannot only be explained by differing levels of risk-taking or impulsivity.

### 4.4. Impulsivity, Risk-Taking, and Decision-Making 

Risk-taking and decision-making under conditions of uncertainty are two behaviors in which the likelihood of outcomes is uncertain, but these differ with regard to decisions about probability [59]. Uncertainty and risky decision are related but distinct concept [44]. Indeed, under risky conditions, subjects can use probability models to evaluate the level of danger, reflected by an increase in the number of pumps on the BART. In the beginning stages of the IGT, information on probability is unknown and unusable until the development of specific knowledge of the IGT. The distinction between risk-taking and impulsivity is also difficult to assess [51]. The link between impulsivity that is measured by psychometric tests and BART scores is still debated [48]. The present study showed that the BART may have several indicators of risk-taking (total score and average score), which is one of the elements that comprises impulsivity. Contrary to some observations, risk-taking as evaluated by the BART should not be considered synonymous with impulsivity. Since a risk-taking is not synonymous of pathological process, several aspects of behavior must be taken into account before hastily concluding a pathological trait in a specific population.

### 4.5. Limitations and Recommendations for Future Works

Although there are considerable conclusions to be considered from this study, limitations must be noted. The main limitation of the present study is the relatively small number of volunteers in our subgroups classified according to their net score at the IGT. In that respect, results on subgroups are exploratory and toned to be confirmed on a larger scale.

Future works should also take into account any possible age and gender differences in decision-making and risk management, which were not considered in our analysis. It is unclear whether gender may have a repercussion on the results of the current study, as men and women tend to perform at the same level on the IGT, although women tend to take additional trials before selecting long-term advantageous decks [60] However, it is worth note that women tend to take less risks than men do in a wide range of behaviors [61]. Age is also an important parameter to take into account, as older adults tend to make more risky decisions relative to younger adults on the IGT, but are more risk averse relative to younger adults on the BART [62]. Future studies should consider how decision-making, risk-taking and impulsivity interact in a pathological population to more clearly define each pathological behavior.

## 5. Conclusions

In conclusion, we propose that the development of correct decision-making under uncertain conditions is mediated by individuals’ predisposition to perceive and orient themselves with their emotions and ability to encode information and use it to explore. In uncertain situations, risk-taking is sometimes necessary to explore the environment and different options that are available. Right/left theta asymmetry may reflect an individual predisposition to be motivated and quickly explore strategies to achieve better performance at the beginning of the BART.

## Figures and Tables

**Figure 1 brainsci-09-00248-f001:**
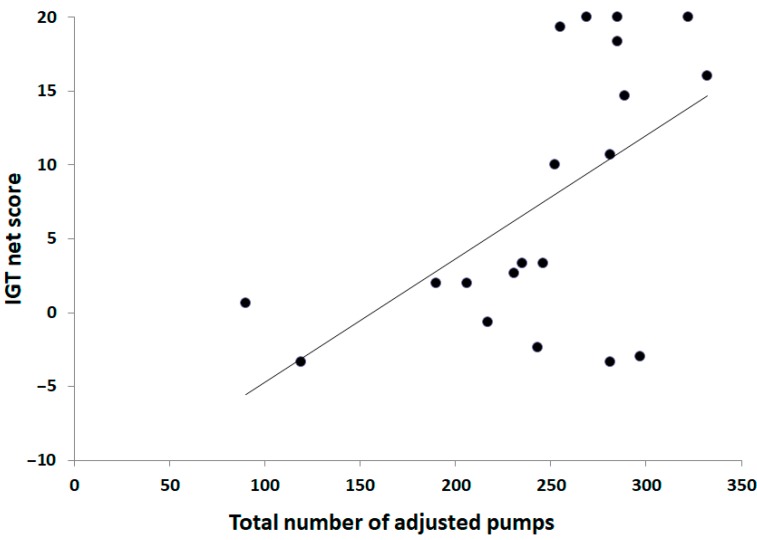
Scatterplot of the relationship between the total number of adjusted pumps in the 10 first trials and the net score on the Iowa Gambling Task (IGT) during the last 60 trials for all participants.

**Figure 2 brainsci-09-00248-f002:**
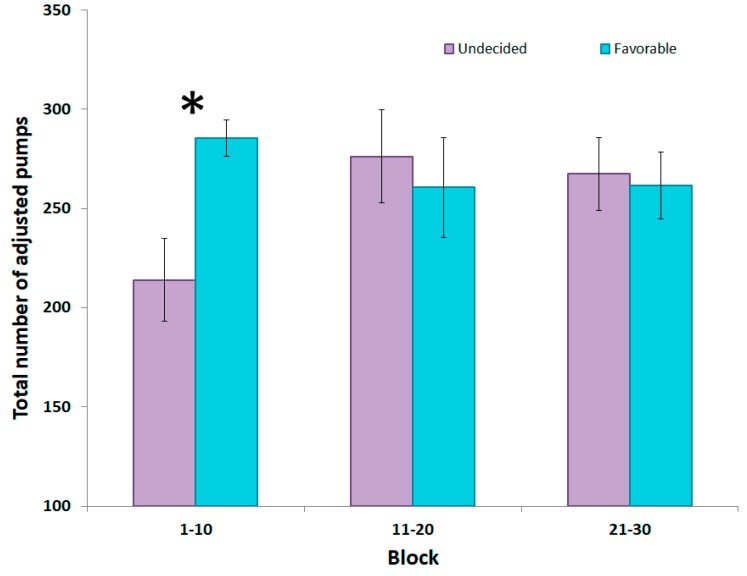
Mean total number of adjusted pumps across the three blocks of the Balloon Analog Risk Task (BART) in the undecided and favorable groups. A significant difference was found between the groups in the first 10 trials. * *p* < 0.05.

**Figure 3 brainsci-09-00248-f003:**
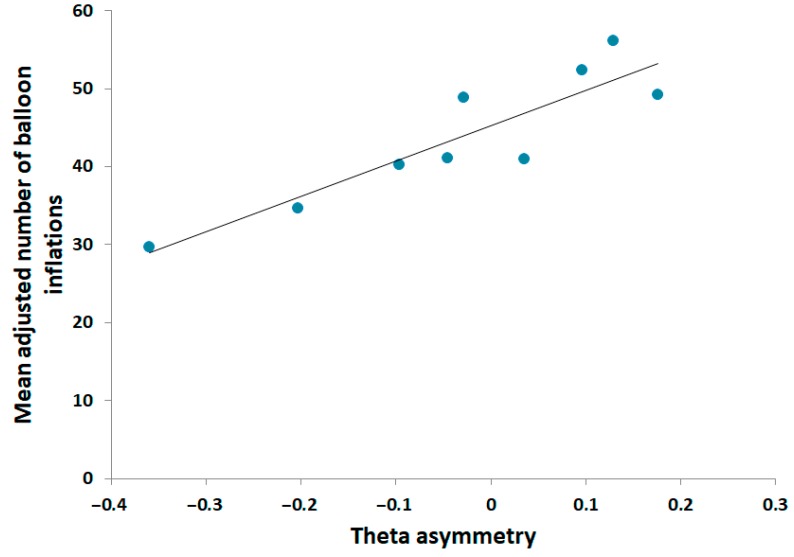
Scatterplot of the relationship between the mean adjusted number of pumps and resting state theta asymmetry in favorable subjects.

**Table 1 brainsci-09-00248-t001:** Psychometric data. Self-report scales were used to measure pathological gambling with the South Oaks Gambling Screen (SOGS) [37], alcohol dependence with the Alcohol Use Disorder Identification Test (AUDIT) [38], the noxious use of cannabis with the Diminuer Entourage Trop Cannabis/Cut-down Annoyed Guilty Eye-opener DETC/CAGE [38], addiction to nicotine with the Fagerström test [38], the existence and intensity of depressive symptoms with the Beck Depression Inventory abbreviated version (BDI) [39], anxiety with the Liebowitz scale [40], impulsivity with the Barratt Impulsiveness Scale version 10 (BIS-10) [42], and individual differences in the five personality dimensions with the Big Five Inventory-French (BFI-Fr) [41].

	Favorable Groups	Undecided Group	Favorable versus Undecided Group	
	Average	Average	*t*-Value	*p*-Value
	(SD)	(SD)		(Two Tailed)
**BDI**	1.11 (0.93)	2 (2.14)	−1.15	0.26
**BFI-Fr**				
Neuroticism	21.33 (5.36)	18.63 (7.03)	0.95	0.36
Extraversion	27.33 (5.70)	26.09 (6.11)	0.47	0.65
Openness to experience	36.33 (6.12)	35.54 (5.64)	0.29	0.77
Agreableness	41.55 (5.08)	42.18 (6.29)	−0.24	0.81
Conscientiousness	33.67 (7.50)	36.45 (4.99)	−0.99	0.33
**LIEBOWITZ SCALE**	6.36 (1.49)	5.20 (1.08)	0.64	0.53
**Performance**	6.05 (1.96)	5.91 (0.77)	0.08	0.93
Anxiety	7.44 (4.27)	6.45 (5.14)	0.46	0.65
Avoidance	4.67 (3.74)	5.36 (4.13)	−0.39	0.7
**Social interaction**	6.67 (1.57)	4.50 (0.96)	1.06	0.3
Anxiety	7.78 (5.47)	5.18 (4.64)	1.15	0.27
Avoidance	5.55 (5.34)	3.81 (4.24)	0.81	0.43
**AUDIT**	4.55 (3.74)	2.36 (2.16)	1.64	0.12
**Fagerström Scale**	0.44 (1.33)	0.18 (0.60)	0.59	0.56
**DETC/CAGE**	0	0	-	-
**SOGS**	0	0.09 (0.30)	−0.9	0.38

**Table 2 brainsci-09-00248-t002:** Spearman’s correlation between BART scores (average and total number of adjusted pumps) and (Barratt Impulsiveness Scale version 10) BIS -10 scores. Spearman *R* values are followed by *t* values in brackets. * *p* < 0.01. When significance was obtained for the overall group of participants, additional significance was analyzed for the favorable and undecided groups.

	BART					
	Average			Total		
	Whole Subjects	Advantageous	Undecided	Whole Subjects	Advantageous	Undecided
	*N* = 20	*N* = 9	*N* = 11	*N* = 20	*N* = 9	*N* = 11
**BIS-10 motor**	−0.16 (−0.69)	-	-	0.39 (1.80)	-	-
**BIS-10 cognitive**	−0.68 (−3.89) *	0.16 (0.44)	−0.68 (−3.59) *	−0.33 (−1.50)	-	-
**BIS-10 no planning**	−0.35 (−1.58)	-	-	−0.27 (−1.20)	-	-
**BIS-10 total**	−0.50 (−2.43)	-	-	−0.03 (−0.14)	-	-

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
