# Peer review of "Behavioral and Electrophysiological Arguments in Favor of a Relationship between Impulsivity, Risk-Taking, and Success on the Iowa Gambling Task"

_brainsci, 2019, doi:10.3390/brainsci9100248_

Round 1

Reviewer 1 Report

Major Concern--The authors do a good job of explaining the relation between performance on the BART, impulsivity,  risk-taking and performance on the IGT in the discussion. The reader would have a greater appreciation of the importance of this work if the authors provided a more in-depth discussion of the aforementioned constructs  in the Introduction. The authors should also focus on the relation between the PFC and the ability to develop a successful strategy in the Introduction.

Minor comments-- In the discussion, the authors indicate that risk-taking on the IGT can only be measured during the last 60 trials.  Actually, there is evidence that participants have implicit knowledge that some selections are risky even though they can't verbalize it.  Usually the knowledge of risk is manifested by "somatic markers" of arousal (e.g., faster heart beat, increased perspiration, etc.). The authors should acknowledge that although individuals may not be explicitly aware of risk, they are frequently implicitly aware of it.

There were a few minor grammatical errors.

Author Response

Comments and Suggestions for Authors

Major Concern--The authors do a good job of explaining the relation between performance on the BART, impulsivity, risk-taking and performance on the IGT in the discussion. The reader would have a greater appreciation of the importance of this work if the authors provided a more in-depth discussion of the aforementioned constructs in the Introduction. The authors should also focus on the relation between the PFC and the ability to develop a successful strategy in the Introduction.

Answer:

In agreement with the reviewer, we have developed in the introduction the strong relationship existing between decision-making, impulsivity, and risk-taking at the lines 49-53.

If the IGT was originally designed to support the somatic marker hypothesis [1,16], several others factors appear to have a major impact on decision-making. Since the IGT was founded on the principle that real-life decision-making involves choices that are based on expected but uncertain rewards and penalties, and that optimal choices are based on well-considered strategies, one can suggest that impulse control problems may alter our decisions.

[…]

And at the lines 64-69:

“To provide a more objective and biologically based measure, the Balloon Analogue Risk Task (BART) was specifically designed to measure the propensity for risk taking [24,25]. Impulsivity is indeed closely related to risk-taking. High levels of impulsivity often result in risky behaviors, such as the initiation of alcohol or drug use [26]. At the same time, almost any human decision carries some risk. Decision making and risk are two inter-related factors, both being related to various uncertainties [27].”

We have also described the relation between the PFC and the ability to develop a successful strategy at the line 85-88:

“Therefore, baseline anterior asymmetries can be used as a predictor of performance on behavioral tasks [29]. The left PFC was previously associated to individual predisposition to respond in terms of approach-related and reward-related motivations and emotions [30–33]. Based on these previous results, we hypothesized that hemispheric asymmetry predicts behavioral profiles with regard to performance on the BART, potentially explaining IGT performance.”

Minor comments-- In the discussion, the authors indicate that risk-taking on the IGT can only be measured during the last 60 trials. Actually, there is evidence that participants have implicit knowledge that some selections are risky even though they can't verbalize it.  Usually the knowledge of risk is manifested by "somatic markers" of arousal (e.g., faster heart beat, increased perspiration, etc.). The authors should acknowledge that although individuals may not be explicitly aware of risk, they are frequently implicitly aware of it.

Answer:

In agreement with the reviewer, we have briefly developed in the discussion the relationship between the somatic marker, risk anticipation and the risk taking. We added the following text at the lines 396-399:

“Somatic markers traduced an unconscious and conscious knowledge of the risk. This is reflected by a change in the autonomic arousal before decks selections [1,7]. However, the risk anticipation does not seem to imply strategy differences and differences in risk-taking in healthy subjects [14]. Furthermore, risk-taking requires a conscious or unconscious knowledge of the risk level [47]. “

Reviewer 2 Report

This study was fairly straightforward and well-executed. Its strength is its simplicity, and that it adds valuable information to the extant literature on the relationship between impulsivity, risk-taking, and decision-making. The study found evidence supporting the advantage of risk-taking in the development of successful decision-making strategies. Frontal theta asymmetry suggested a neural underpinning to the development of these strategies.
The authors provide a clear and concise account of the relevant literature and study findings. There are some comments that should be addressed by the authors.

1) Lines 65-69: “Changes in decisions are frequently justified by an important feeling, led by impulsivity or risk taking with a lack of distinction of how each of these constructs specifically influences behaviors and decisions. However, in clinical practice, an alteration in decision-making could have various causes, which complicates providing appropriate care.”
The meaning of this section is unclear. Is it related to the somatic marker hypothesis? Need to clarify the assertion made here.

2) “All of the balloons had a different explosion threshold”: what was the range of the possible number of pumps?

3) Lines 198-199: “A significant correlation was found between the net score on the IGT during the last 60 trials and the total adjusted number of pumps in the first 10 trials of the BART…”
What was the rationale for correlating the first block of trials on the BART with the last 60 trials of the IGT? The rationale for using the last 60 trials of the IGT, that the “participants had started developing a strategy”, was reasonable. But, the same might be said for the BART - that the participants were developing a strategy during the initial trials. You did find a difference in performance on the first 10 trials compared to the later trials (lines 195-196). If so, then why perform the comparison you did? It might be more informative to correlate the later trials from both tasks when strategies have “stabilized”.   

4) Lines 211-212: “The group who developed a favorable strategy showed more risk taking, i.e. more presses, in the first 10 trials of the BART”
Based on Figure 2, it seems that it was the Undecided group that showed less risk-taking in the first 10 BART trials (rather than more in the Favorable group). There was no within-group comparisons between blocks, but it looks like there is likely not a difference across blocks for the Favorable group (thus, their strategy may not have changed throughout the task), while block 1 (the first 10 trials) may differ from later trials for the Undecided group. This difference may require adjusting conclusions of the study.

5) In Table 2 the groups are referred to as ‘Advantageous’ and ‘Undecided’, while elsewhere the groups are referred to as ‘Favorable’ and ‘Undecided’. Need to reconcile how the groups are referred to.

6) Line 256: “No significant relationship was”
Incomplete sentence

7) Line 258-259: Should avoid the use of word “influence” (here, and throughout Discussion), which implies causation or directionality. Conclusions are based on correlational analyses.

8) Discussion section 4.1: The discussion of the relationship between risk-taking and impulsivity is framed as the favorable group making riskier decisions at the beginning of the BART, which “accelerated the learning process”. However, the data suggest that it may be the undecided group that is performing differently, by making less risky decisions at the beginning of the BART (see my comment 4 above). While this might not necessarily change the conclusions of the study, the authors should base their conclusions on the data. This theme is present throughout the Discussion section, and should be addressed accordingly.

9) Discussion of limitations was well-considered. Especially the potential influence of Age on outcomes, as there was a relatively large range of ages included in the fairly small sample of subjects.

Author Response

Comments and Suggestions for Authors

This study was fairly straightforward and well-executed. Its strength is its simplicity, and that it adds valuable information to the extant literature on the relationship between impulsivity, risk-taking, and decision-making. The study found evidence supporting the advantage of risk-taking in the development of successful decision-making strategies. Frontal theta asymmetry suggested a neural underpinning to the development of these strategies. The authors provide a clear and concise account of the relevant literature and study findings. There are some comments that should be addressed by the authors.

1) Lines 65-69: “Changes in decisions are frequently justified by an important feeling, led by impulsivity or risk taking with a lack of distinction of how each of these constructs specifically influences behaviors and decisions. However, in clinical practice, an alteration in decision-making could have various causes, which complicates providing appropriate care.” The meaning of this section is unclear. Is it related to the somatic marker hypothesis? Need to clarify the assertion made here.

Answer 1: Reviewer 1 requested a text modification to better explain and clarify the assertion. By “feeling” we were indeed thinking of the somatic marker hypothesis, and it is true that the word feeling was not the most appropriate. In agreement with these comments, we modified the text at the lines 49-50:

If the IGT was originally designed to support the somatic marker hypothesis [1,16], several others factors appear to have a major impact on decision-making.

And at the lines 73-78:

Changes in decisions are frequently justified by an important feeling, led by impulsivity or risk taking with a lack of distinction of how each of these constructs specifically influences behaviors and decisions. Indeed, the somatic marker hypothesis refers to the collection of signals related to the body and the brain that characterize the emotional and affective responses [29]. In this sense, the impact of impulsivity or risk-taking on decision-making does not contradict the somatic marker hypothesis.

2) “All of the balloons had a different explosion threshold”: what was the range of the possible number of pumps?

Answer 2:

The number of pumps is in line with the original version described by Lejuez. Balloon breakpoints range from 1 to 128 pumps. For example, for a 1-128 breakpoint, the probability that a balloon would explode on the first pump is 1/128. If the balloon did not explode on the first pump, the probability that the balloon would explode on the second pump is 1/127, and so on. This information has been added in the text at the lines 170-171.

All of the balloons had a different explosion threshold. Balloon breakpoints ranged from 1 to 128 pumps.

3) Lines 198-199: “A significant correlation was found between the net score on the IGT during the last 60 trials and the total adjusted number of pumps in the first 10 trials of the BART…”
What was the rationale for correlating the first block of trials on the BART with the last 60 trials of the IGT? The rationale for using the last 60 trials of the IGT, that the “participants had started developing a strategy”, was reasonable. But, the same might be said for the BART - that the participants were developing a strategy during the initial trials. You did find a difference in performance on the first 10 trials compared to the later trials (lines 195-196). If so, then why perform the comparison you did? It might be more informative to correlate the later trials from both tasks when strategies have “stabilized”. 

Answer 3:

To answer your comment, we added in the result section the text lines 214-217:

To study whether the reduced performances in the first 10 trials could be imputable to the development of an efficient strategy, we explored their relationship with IGT performances. The first trials of the BART display an uncertainty level which confers a structure close to the last trials of the IGT [47].

We also added the following text in the discussion section at lines 325-332:

To understand this association between the later stage of the IGT and BART scores, risk taking in the early and late stages of the IGT should be considered separately [48]. This is different from the BART, which was specifically designed to measure the propensity for risk taking from the beginning of the task [24,25]. Nevertheless, the correlation observed between the IGT and the first block of the BART is not inconsistent with the literature. During the BART, participants did not know about the probability distribution of explosions which confers an uncertainty level at the task in addition to the risk, mostly on the first trials [47]. In this context, the first trials of the BART share similar mechanisms as the last trials of the IGT. »

We also added the statistical analysis made on the others blocks, which all showed a lack of correlation. this additional information was made at the lines 220-223:

Furthermore, no relationship was found between the IGT net score during the last 60 trials of the IGT and the total adjusted number of pumps in the trials 11-20 (Spearman R = -0.03, t18 = -0.15, p = 0.88) and 21-30 (Spearman R = -0.02, t18 = -0.07, p = 0.94) of the BART.

4) Lines 211-212: “The group who developed a favorable strategy showed more risk taking, i.e. more presses, in the first 10 trials of the BART” Based on Figure 2, it seems that it was the Undecided group that showed less risk-taking in the first 10 BART trials (rather than more in the Favorable group). There was no within-group comparisons between blocks, but it looks like there is likely not a difference across blocks for the Favorable group (thus, their strategy may not have changed throughout the task), while block 1 (the first 10 trials) may differ from later trials for the Undecided group. This difference may require adjusting conclusions of the study.

Answer 4:

The reviewer is right, both interpretations are possible. Following the reviewer suggestion, we analyzed whether there was within-group differences between blocks. It appears that the undecided group shows a difference of performances in the first block, compared to the two consecutive blocks. These results have been added to the result section (lines 232-236):

“A significant block x group interaction (F2,36 = 6.2811, p < 0.01; Figure 2) was observed. The group who failed to develop a favorable strategy showed less risk taking, i.e. fewer presses compared to the Favorable group in the first 10 trials of the BART (Fisher LSD test, p<0.05). Risk-taking of the undecided group was also reduced in the first 10 trials compared to trials 11-20 and trials 21-30 (p<0.01 for both comparisons).“

Basically, these results do not change fundamentally our conclusion. We nevertheless made a few changes in the discussion to explain both interpretations at lines 295-316.

Differences in impulsivity and risk taking were found between participants who were able to develop a correct strategy and those who could not. The undecided group made significantly fewer pumps than the favorable group in the first block of the BART. The undecided group was more cautious than the favorable group at the beginning of the task, even if both groups had the same level of performances in the following blocks. Compared to the undecided group, favorable group took more risk at the beginning of the task, to achieve more rapidly a high level of performance.”

[…]

And at the lines 320-324:

« Our initial hypothesis that we would observe increased risk taking in the undecided group was not confirmed. It was actually quite the opposite that was observed, the undecided group showing a reduced risk propensity at the beginning of the BART. The subjects in the undecided group were unable to develop the appropriate strategies and consequently did not consciously take risks [28]. »

5) In Table 2 the groups are referred to as ‘Advantageous’ and ‘Undecided’, while elsewhere the groups are referred to as ‘Favorable’ and ‘Undecided’. Need to reconcile how the groups are referred to.

Answer 5:

Thank you for your conscientious reading. We modified the table with the correct term “favorable”.

6) Line 256: “No significant relationship was” Incomplete sentence

Answer 6:

Again, thank you for your conscientious reading. We added the missing part at the lines 280-281. No significant relationship was found between BIS-10 global and subscale scores and theta asymmetry (p > 0.1).

7) Line 258-259: Should avoid the use of word “influence” (here, and throughout Discussion), which implies causation or directionality. Conclusions are based on correlational analyses.

Answer 7:

We apologize for the incorrect use of the word “influence” that is not the most appropriate term as you said. We modified the sentence lines 284-285. Several changes have also been made throughout the discussion.

The present study explored the role of impulsivity and risk-taking on decision-making performance on the IGT and the impact of frontal lobe asymmetry on decision making.”

8) Discussion section 4.1: The discussion of the relationship between risk-taking and impulsivity is framed as the favorable group making riskier decisions at the beginning of the BART, which “accelerated the learning process”. However, the data suggest that it may be the undecided group that is performing differently, by making less risky decisions at the beginning of the BART (see my comment 4 above). While this might not necessarily change the conclusions of the study, the authors should base their conclusions on the data. This theme is present throughout the Discussion section, and should be addressed accordingly.

Answer 8:

Your comments showed that our explanation was lacking of clarity. Thanks to your advice, we modified the manuscript lines 295-316.

Differences in impulsivity and risk taking were found between participants who were able to develop a correct strategy and those who could not. The undecided group made significantly fewer pumps than the favorable group in the first block of the BART. The undecided group was more cautious than the favorable group at the beginning of the task, even if both groups had the same level of performances in the following blocks. Compared to the undecided group, favorable group took more risk at the beginning of the task, to achieve more rapidly a high level of performance.”

[…]

And at the lines 320-324:

“Our initial hypothesis that we would observe increased risk taking in the undecided group was not confirmed. It was actually quite the opposite that was observed, the undecided group showing a reduced risk propensity at the beginning of the BART. The subjects in the undecided group were unable to develop the appropriate strategies and consequently did not consciously take risks [28].

 […]

And at the lines 329-332:

« These risks accelerated the learning process, thereby allowing the favorable group to earn more money at the start of the game. Since subjects of the undecided group showed a reduced propensity to take risk in the different tasks, they consequently learned more slowly the probability of explosions at the BART and decks composition at the IGT. »

9) Discussion of limitations was well-considered. Especially the potential influence of Age on outcomes, as there was a relatively large range of ages included in the fairly small sample of subjects.

Reviewer 3 Report

The  study investigated the relationship between trait  impulsivity, risk taking, and decision-making performance.  The impulsivity was measured by the Barratt Impulsiveness Scale. Resting state neural activity was recorded to explore whether brain oscillatory rhythms provide important information about the dispositional trait of impulsivity. 

I think that the idea can be fine, but it is not well developed

_major points: 1)Please, explaine better the EEG condition. In the paper is not clear what was the instruction during resting state.

2) please, describe better the impedence values and methodological aspects of eeg recording.

3) in the text there is not any figure related the EEG (delta or alpha and so on..), Please, insert some imaging data EEG related

Minor point: The title is really long and not really focused on EEG data too. 

Author Response

The study investigated the relationship between trait impulsivity, risk taking, and decision-making performance. The impulsivity was measured by the Barratt Impulsiveness Scale. Resting state neural activity was recorded to explore whether brain oscillatory rhythms provide important information about the dispositional trait of impulsivity. 

I think that the idea can be fine, but it is not well developed_major points:

1)Please, explaine better the EEG condition. In the paper is not clear what was the instruction during resting state.

Answer 1:

The reviewer rightly pointed out that some information was missing in the manuscript. We added the following information at the lines 138-139:

“An EEG recording at rest was performed immediately after the IGT. During the recording, subjects were instructed to close their eyes and remain still”.

2) please, describe better the impedence values and methodological aspects of EEG recording.

Answer 2:

The reviewer is right, some information was missing about the EEG recording:

Consistent with recommended acquisition protocols for high-impedance EEG systems (Electrical Geodesic Inc., Eugene, OR), the target impedance level for the electrodes was below 30kΩ during data collection. We added in the text at the lines 143-144:

“All electrode impedances were below 30 kΩ before recording was started”

In the text, we added at the lines 147-148: “Raw EEG data were offline re-referenced to a common average” Bad channels. In the text we added the following information at the lines 148-149: “Channels with excessive noise (due to malfunctioning or bad signal during data collection) were replaced using topographic interpolation.” In the text we added that “Each artifact-free period was further segmented into epochs of 2 seconds” at the lines 150-151.

3) in the text there is not any figure related the EEG (delta or alpha and so on..), Please, insert some imaging data EEG related

Answer 3:

We understand that the reviewer would like to see the figure of an EEG recording, and we can do it easily, but we do not see what kind of figure would be appropriate. Since EEG data needs to be processed before extracting the parameters of interest, visual raw data appears not meaningful. When looking at the other EEG articles dealing with a similar topic (for example, references 56 and 57 of the article), these articles do not present any figure of EEG data either. Maybe the reviewer has a specific idea of figure in mind?

Minor point: The title is really long and not really focused on EEG data too. 

Answer:

The title has been changed to “Behavioral and electrophysiological arguments in favor of a relationship between impulsivity, risk taking, and success on the Iowa Gambling Task”